# Assessing the Benefits and Costs of the Hydrogen Cyanide Antiherbivore Defense in *Trifolium repens*

**DOI:** 10.3390/plants12061213

**Published:** 2023-03-07

**Authors:** Hind Emad Fadoul, Lucas J. Albano, Matthew E. Bergman, Michael A. Phillips, Marc T. J. Johnson

**Affiliations:** 1Department of Biology, University of Toronto Mississauga, Mississauga, ON L5L 1C6, Canada; 2Department of Ecology and Evolutionary Biology, University of Toronto, Toronto, ON M5S 3B2, Canada; 3Department of Cell and Systems Biology, University of Toronto, Toronto, ON M5S 3G5, Canada

**Keywords:** autotoxicity, cyanogenesis, freezing tolerance, plant defense, plant-herbivore, trade-off

## Abstract

Understanding the evolution of plant defenses against herbivores requires identifying the benefits and costs of defense. Here, we tested the hypothesis that the benefits and costs of hydrogen cyanide (HCN) defense against herbivory on white clover (*Trifolium repens*) are temperature dependent. We first tested how temperature affected HCN production in vitro, and then examined how temperature influenced the efficacy of HCN defense of *T. repens* against a generalist slug (*Deroceras reticulatum*) herbivore using no-choice and choice feeding trial assays. To understand how temperature affected the costs of defense, plants were exposed to freezing, and HCN production, photosynthetic activity, and ATP concentration were quantified. HCN production increased linearly from 5 °C to 50 °C, and cyanogenic plants experienced reduced herbivory compared to acyanogenic plants only at warmer temperatures when fed upon by young slugs. Freezing temperatures induced cyanogenesis in *T. repens* and decreased chlorophyll fluorescence. Cyanogenic plants experienced lower ATP levels than acyanogenic plants due to freezing. Our study provides evidence that the benefits of HCN defense against herbivores are temperature dependent, and freezing may inhibit ATP production in cyanogenic plants, but the physiological performance of all plants recovered quickly following short-term freezing. These results contribute to understanding how varying environments alter the benefits and costs of defense in a model system for the study of plant chemical defenses against herbivores.

## 1. Introduction

Plants produce a large variety of secondary metabolites that have evolved as adaptations to mitigate abiotic and biotic stresses, including chemical defenses against herbivores [1]. Understanding the evolution of plant defenses against herbivores requires identifying and quantifying the benefits and costs of defense [2]. The direct benefits of anti-herbivore defenses depend on the degree to which a defense increases the fitness of a plant in the presence of herbivores [3]. However, defenses are also expected to have costs. For example, virtually all defenses have a cost in terms of resource allocation associated with the synthesis, storage, and regulation of plant secondary metabolites [4,5,6]. These allocation costs reduce fitness in the absence of herbivores when comparing plants with a particular defense versus plants without the defense [7]. However, costs can also be expressed when a defense has pleiotropic effects, such as reducing a plant’s tolerance to abiotic stresses [8,9]. Therefore, in assessing the costs and benefits of defense, it is important to recognize that the effects on fitness may change with the environmental context [10].

The importance of environmental context for the benefits and costs of defense has been studied extensively at a macroevolutionary scale [11,12]. By contrast, how environmental variation affects the costs and benefits of defense within species, and thus the microevolution of defenses, has received less attention [13,14]. Here we consider how the benefits and costs of the chemical defense hydrogen cyanide (HCN) depend on variation in temperature. HCN is produced by approximately 3000 different plant species from over 130 plant families [15,16]. HCN is released from plant tissues through the enzymatic reaction of beta-glucosidase (e.g., linamarase) with cyanogenic glycosides (e.g., lotaustralin and linamarin) following tissue damage, which cleaves the sugar moiety from the glycoside to liberate highly toxic HCN [16]. When ingested, HCN inhibits cytochrome oxidase in the electron transport chain [17], and thus it negatively affects the mitochondrial respiration pathway responsible for the production of ATP. Previous research suggested that the benefits of HCN and its fitness costs may depend on variation in temperature in both space and time [18,19], but recent research [20] has called these earlier results into question. We sought to provide some insight into the contradictory conclusions of previous studies. 

White clover (*Trifolium repens*) is a model for the study of HCN defense evolution. White clover plants are polymorphic for cyanogenesis, in that some plants produce HCN (i.e., cyanogenic) and others lack the ability to produce HCN (i.e., acyanogenic) [16,21]. The polymorphism arises through two independently segregating loci [22,23,24,25]. The first locus (hereafter referred to as *Ac*) is a three-gene region that includes the rate-limiting enzyme CYP79D15, which is required for the production of cyanogenic glycosides [26]. The *Ac* locus has both a dominant functional allele (*Ac*) and a recessive nonfunctional deletion allele (*ac*), which results in the presence and absence of cyanogenic glycosides, respectively. The second locus *Li* encodes for linamarase, which hydrolyzes cyanogenic glycosides to produce HCN. *Li* also has a dominant functional allele (*Li*) and a recessive nonfunctional deletion allele (*li*), which determines the presence or absence of linamarase, respectively. A plant needs at least one dominant allele at both *Ac* and *Li* to produce HCN following tissue damage, while plants that are homozygous recessive at either locus are acyanogenic [16,23,24,27].

The frequency of HCN, *Ac/ac*, and *Li/li* within populations of *T. repens* varies spatially over large and small scales [28,29,30,31]. Multiple biotic and abiotic factors have been proposed as potential selective agents to explain this spatial variation in cyanogenesis and its underlying allele frequencies. These factors include variation in herbivore pressure [16,32,33], drought and water stress [34,35,36], and soil nutrient levels [37]. However, the earliest and most strongly supported hypothesis is that HCN, *Ac*, and/or *Li* decrease the tolerance of plants to freezing temperatures [18,31]. The relationship between temperature and the frequency of HCN, *Ac*, and *Li* within populations was first noted by Daday [29,30], who found that the *Ac* and *Li* alleles were most common in warmer climates (i.e., low latitudes and low elevations), whereas *ac* and *li* were most common in colder climates. This observation has been supported by multiple studies [35,38,39]. Daday was also the first to report evidence that cyanogenic plants incurred more damage following freezing [18], which led to the hypothesis that HCN is autotoxic for plants and selected against in regions with freezing temperatures [18,31]. While this hypothesis has received experimental support [40,41,42], a recent experimental study found that cyanogenic plants did not exhibit decreased physiological performance or growth compared to acyanogenic plants [20]. However, acyanogenic plants that contained a functional *Ac* or *Li* allele exhibited decreased survival and increased tissue damage following freezing compared to plants homozygous for *ac* and *li*, suggesting that investment in cyanogenic glycosides and linamarase may impose allocation costs that are exacerbated by freezing, as opposed to cyanogenesis being physiologically autotoxic following freezing. A limitation of all previous studies is that they did not directly measure the effect of HCN on ATP production following freezing, which is the physiological mechanism by which HCN is predicted to cause autotoxicity [18].

In this study, we tested the hypothesis that both the benefits and costs of HCN are dependent on temperature. Specifically, we investigated two hypotheses: (1) The benefits of HCN defense are dependent on temperature, whereby HCN production is predicted to increase as temperatures increase above freezing. We tested this hypothesis by assessing how the production of HCN varied with temperature, and whether its efficacy as a defense against herbivores was temperature dependent. (2) HCN imposes a physiological cost through autotoxicity of plant tissues following freezing damage. We predicted that freezing would lead to increased production of HCN in plant tissues, HCN-producing plants will exhibit reduced ATP production following freezing, and HCN-producing plants will experience impaired physiological responses (i.e., photosynthesis) following freezing. We tested these hypotheses using a series of in vitro and whole plant experiments in which we experimentally randomized *Ac*/*ac* and *Li*/*li* alleles on a common genetic background. Our results contribute to understanding how varying environments alter the benefits and costs of defense in a model system for the study of plant chemical defenses against herbivores. 

## 2. Results

### 2.1. Effect of Temperature on HCN Production

HCN production was highly dependent on temperature. The quantity of HCN produced increased linearly with temperature at a rate of 0.26 μmol HCN per 1 °C (Figure 1; F_9,20_ = 131.89, *p* < 0.001). While a linear curve was the best fit to the data, HCN production increased by just 1.3× from 5 °C to 15 °C, whereas it increased by 1.9× from 15 °C to 25 °C, suggesting HCN production increases more rapidly at temperatures > 15 °C (Figure 1). 

### 2.2. Effect of Temperature on Efficacy of HCN Defense against Herbivores

#### 2.2.1. No-Choice Assay

We found evidence that the efficacy of HCN against herbivores varied with temperature, but only in the first trial when slugs were at an earlier developmental stage. In trial 1, herbivory was 41% lower on cyanogenic (i.e., AcLi) than acyanogenic plants (i.e., Acli, acLi, acli), there was no main effect of temperature, and there was a marginally non-significant cyanotype × temperature interaction (Table 1). Cyanogenic plants experienced 85% less herbivory than acyanogenic plants at 25 °C (t_72_ = 2.848, *p* = 0.006), but 8% more herbivory at 15 °C (t_72_ = 0.498, *p* = 0.620). When leaves that received no herbivory were excluded from analysis, the effect of cyanotype (F_3,56_ = 6.282, *p* < 0.001) and cyanotype × temperature (F_3,56_ = 3.852, *p* = 0.014) were even stronger (Appendix A), and the qualitative and quantitative trends were similar to those described above (Appendix A). In trial 2, when slugs were larger and at a later development stage, cyanogenic plants experienced 34% less herbivory, but there was no significant effect of cyanotype, temperature, or cyanotype × temperature interaction (Table 1, Appendix A). Results that only considered leaves that received damage (Appendix A) or considered the binomial response of whether or not leaves were consumed (data not shown) led to similar conclusions. 

#### 2.2.2. Choice Assay

In choice assays, cyanogenic plants experienced 71% lower herbivory, but the effect of cyanotype was not significant, and there was no effect of temperature or cyanotype × temperature interaction (Table 1, Figure 2). Results that only considered leaves that received damage (Appendix A) or considered the binomial response of whether or not leaves were consumed (data not shown) led to similar conclusions.

### 2.3. Freezing and Autotoxicity

#### 2.3.1. HCN Quantification

HCN quantification results for AcLi plants showed that HCN concentration significantly increased by 2.5 × 3 h after freezing (t_40_ = 5.443, *p* < 0.001) and was still elevated by 2× compared to controls 24 h after freezing (t_40_ = 3.450, *p* = 0.001). HCN concentration returned close to pre-freezing levels within 7 days after freezing (Figure 3). 

#### 2.3.2. Analysis of Photosynthetic Activity Fv/Fm

Freezing inhibited photosynthetic activity, but this effect did not vary among cyanotypes (F_3,190_ = 1.396, *p* = 0.245). Leaf photosystem II Fv/Fm varied through time (F_4,190_ = 24.443, *p* = < 0.001, Appendix A), such that there was a 5% decrease in Fv/Fm during cold acclimation compared to the control period at ambient temperature, and this depression of photosynthesis continued 3 h after freezing. Fv/Fm then increased to pre-freezing “control” levels within 24 h after freezing. There was no cyanotype × time point interaction (F_12_,_190_ = 1.169, *p* = 0.308). 

#### 2.3.3. ATP Production

ATP concentrations in leaves differed among cyanotypes, and this effect varied before and after freezing (Table 2, Figure 4). Among control plants, cyanogenic plants produced 59% lower ATP concentrations than acyanogenic plants (t_121_ = 3.158, *p* = 0.002). Cyanogenic plants produced 92% lower ATP concentrations than acyanogenic plants 3 h after freezing (t_121_ = 6.513, *p* < 0.001), such that the reduction in ATP in cyanogenic plants compared to acyanogenic plants was 2.5× higher 3 h after freezing compared to before freezing (t_121_ = 2.477, *p* = 0.015). Twenty-four hours after freezing, cyanogenic plants produced 49% lower ATP concentrations than acyanogenic plants (t_121_ = 2.362, *p* = 0.020), which was statistically similar to the reduction seen in control plants (t_121_ = 0.573, *p* = 0.568). After 7d post-freezing, there was no difference in ATP concentrations between cyanogenic and acyanogenic plants (t_121_ = 0.404, *p* = 0.687).

## 3. Discussion

Plant cyanogenesis is a potent defense against a wide range of both generalist and specialist insect herbivores [43,44]. Surprisingly, the physiological effects and evolution of this chemical defense are still not fully understood after over a century of research [21,29,35,45,46]. Here we tested the hypothesis that both the benefits and costs of HCN are dependent on temperature. Our findings indicated that HCN production is highly dependent on temperature, with the concentration and efficacy of defense increasing with warmer temperatures and under freezing temperatures, inhibiting a key physiological process—ATP production. These results provide novel insight into the benefits and costs shaping the evolution of cyanogenesis in *Trifolium repens* specifically, and the evolution of plant defenses more generally.

### 3.1. Effect of Different Temperatures on HCN Production and Efficacy against Herbivores

Our results demonstrate that the HCN chemical defense is highly dependent on temperature. The rate of hydrolysis of cyanogenic glycosides (i.e., lotaustralin and linamarin) by linamarase increased with temperature. Specifically, HCN production was 87% greater at 25 °C than at 15 °C (Figure 1), which led to enhanced resistance against slug herbivory (Figure 2). Our findings add to previous results showing that HCN defenses and expression of the underlying metabolic components depend on temperature. For example, Collinge and Hughes [47] showed that cyanogenic glycoside content in three individuals of *T. repens* was maximized at moderate constant temperatures (above 18 °C) and decreased at lower temperatures. Hayden and Parker [19] showed a significant reduction of HCN expression in cyanogenic *T. repens* at cold temperatures (15 °C/5 °C day/night) compared to warmer temperatures (25 °C/15 °C day/night). Ikya et al. [48] found that linamarase activity increased with temperature from 25 °C to 35 °C. Our results show that even when the concentrations of cyanogenic glycosides and linamarase are kept constant, the kinetics of hydrolysis of glycosides are temperature dependent, such that HCN production increases steadily with increasing temperatures. These results lead to the prediction that the efficacy of HCN as an antiherbivore defense will increase with temperature and may be relatively ineffective at ≤15 °C.

Our herbivore assays are partially consistent with the prediction that HCN defense increases in effectiveness with increasing temperature. In no-choice assays, cyanogenic plants were more resistant to young slugs than acyanogenic plants at 25 °C, and this difference in resistance was not apparent at 15 °C. However, there was no effect of cyanogenesis on resistance to slugs when the slugs were larger (no-choice trial 2) or when slugs were given a choice between leaves of different cyanotypes. Smaller immature slugs are likely more sensitive to HCN than larger adult slugs, which may have a better ability to detoxify and tolerate HCN [33,49]. We may have observed an even greater effect of the temperature dependency of cyanogenesis if the disparity in temperatures was increased, but we avoided such treatments to minimize the confounding effects of temperature on the physiological performance of plants and slugs. Future research may benefit from considering how a wider range of temperatures affects HCN defenses against diverse herbivores. We anticipated the strongest effects of temperature-dependent resistance due to cyanogenesis in the choice assays, but the lack of an effect in this assay may have been caused by slugs avoiding all plants once a cyanogenic plant was detected, since they may not be able to recognize cyanogenic forms without first sampling them [32,50,51]. This phenomenon of associational resistance is well known from other systems [52,53].

### 3.2. Freezing and Autotoxicity

We tested key criteria of the hypothesis that cyanogenesis imposes a physiological cost following freezing damage due to HCN autotoxicity [18,31]. HCN is toxic when it binds to cytochrome oxidase, inhibiting the electron transport chain and preventing ATP production [54]. Several previous studies examined how freezing differentially affects growth, cell damage, survival, and photosynthesis of cyanogenic and acyanogenic plants, with mixed results [18,20,40,41,42]. However, we are unaware of any study that has measured the direct effect of cyanogenesis on ATP production, which, given the mechanism of toxicity, is critical evidence to evaluate whether or not HCN can be autotoxic to plants following freezing. HCN would not be autotoxic if the β-cyanoalanine synthesis pathway rapidly detoxifies HCN in cells following freezing [25,55]. Our results clearly indicate that ATP production was inhibited in cyanogenic morphs, but not in acyanogenic morphs, following freezing stress. ATP production had started to recover within 24 h, but was still lower than acyanogenic plants, whereas it was equivalent to acyanogenic plants within 7 days, and higher than pre-freezing levels, potentially indicative of active cellular repair [56,57,58]. These results indicate that HCN inhibits ATP production in cyanogenic plants after freezing, satisfying a necessary condition for the autotoxicity hypothesis. 

Despite HCN inhibiting ATP following freezing, we saw no evidence of autotoxic effects on photosynthetic activity. We measured chlorophyll fluorescence (i.e., Fv/Fm) as an indicator of photosynthetic efficiency and plant stress after freezing [59]. Fv/Fm is a sensitive indicator of plant photosynthetic performance, with lower values indicating potential stress. While photosynthetic activity decreased in response to freezing, it recovered within 24 h of freezing to pre-freezing levels, and importantly there was no differential response among cyanotypes. These findings are aligned with Kooyers et al. [20], who also found no significant differences among *T. repens* cyanotypes in Fv/Fm following freezing stress. However, one study that exposed *T. repens* and *Lotus corniculatus* to a longer period (18 h) of cold temperatures (1 °C) found that photosynthesis was lower in cyanogenic than acyanogenic plants [41]. Moreover, multiple studies have found that cyanogenic plants experience greater frost damage and lower survival compared to acyanogenic plants [40], although such reduced tolerance to freezing could be due to allocation costs and/or autotoxicity [20]. 

We believe the inconsistent results from existing studies testing the autotoxicity hypothesis can be reconciled. Our results show that cyanogenesis inhibits ATP production following freezing, but multiple studies show inconsistent effects of cyanogenesis on photosynthesis following freezing, while cyanogenesis is frequently associated with greater cellular damage and decreased survival as described above (but see [20]). A limitation of most previous studies of the autotoxicity hypothesis, including our own, is that plants are only exposed to freezing for short periods of time. The effects of repeated exposure to frost and extended periods of freezing could lead to much stronger autotoxic effects than detected in past studies, which may be cumulative with prolonged exposure. Such studies may help explain the physiological mechanisms by which selection leads to the repeated evolution of decreased frequency of cyanogenesis at high latitudes and high altitudes. Moreover, these physiological costs of cyanogenesis are likely to be greatest for plants exhibiting the highest expression of HCN, which is rarely taken into account.

## 4. Conclusions

Our study suggests that both the benefits and costs of the HCN chemical defense vary with temperature. The production of HCN and its efficacy provide evidence that the HCN defense is temperature dependent. HCN only reduced herbivory at temperatures > 15 °C, and our results suggest that HCN concentration will increase steadily with increasing temperature. Our results also suggest that HCN can inhibit ATP production following freezing, meeting a necessary condition of the autotoxicity hypothesis. Based on our results, combined with previous literature, we predict that autotoxicity will increase with prolonged exposure to freezing, which will compound recently documented allocation costs of the metabolic components of the HCN defense (i.e., cyanogenic glycosides and linamarase), with these costs elevated under stress [20]. In putting these results together, we believe this changes the expectations under which cyanogenic genotypes will have higher fitness than acyanogenic genotypes, versus when acyanogenic genotypes are expected to exhibit higher relative fitness (Figure 5). In the absence of temperature-dependent defense expression or autotoxicity, cyanogenic genotypes are expected to evolve more easily, over a wider range of temperatures. With temperature-dependent HCN production and autotoxicity, the evolution of cyanogenesis becomes more restrictive, because the cumulative costs of producing the defense, autotoxicity, and herbivory compound, while the benefits of the HCN defense are only realized in warmer temperatures. These results might help to explain the conditions under which HCN will evolve, especially under changing climates. 

More generally, our results contribute to understanding how environmental variation influences the microevolution of plant defenses. Variation in the environment, and particularly variation in resource availability, is well known to influence the macroevolution of defense by modulating the allocation costs of defense [11]. How variation in the environment influences the microevolution of defense is poorly known [13], and the effects of variation in the environment on the benefits of defense have mostly considered variation in the levels and types of herbivory [60]. Our results suggest that variation in edaphic factors not directly related to resource availability or herbivory can modulate both the costs and benefits of defense. We propose that a more complete understanding of how the environment modulates these benefits and costs of defense would improve our ability to predict the evolution of defenses within species.

## 5. Materials and Methods

### 5.1. Study System

White clover (*Trifolium repens* L., Fabaceae) is a short-lived perennial herbaceous plant, native to Europe and central Asia that is cultivated globally in temperate zones for forage and soil nutrification [62]. The cyanogenesis polymorphism of white clover results in four cyanotypes: cyanogenic plants with a minimum of one dominant allele at each locus (i.e., Ac– Li–) produce both precursors (i.e., cyanogenic glycosides and linamarase) and release HCN, whereas acyanogenic plants include three cyanotypes that produce cyanogenic glycosides but no linamarase (Acli), linamarase but no cyanogenic glycosides (acLi), or neither component (acli). 

#### Plant Materials

To assess the costs and benefits of HCN defense, plants of all four cyanotypes (i.e., AcLi, Acli, acLi, acli) of T. repens were grown from F4 seed originating from an F0 parental cross between an acaclili plant from Ontario, Canada, and an AcAcLiLi plant from Louisiana, USA. The original parental cross resulted in plants that were all cyanogenic and therefore heterozygous AcacLili. These heterozygotes were randomly crossed with one another to produce an F2 population segregating at Ac and Li. Random crosses in the F2 and F3 generations were performed among plants within the same cyanotype (i.e., AcLi, Acli, acLi, acli), which increased the frequency of homozygous genotypes at Ac and Li on a randomized Ontario × Louisiana genetic background. This method allowed us to randomize the Ac/ac and Li/li alleles onto a common recombined and segregating genetic background, and therefore to determine the effects of the Ac and Li loci on the costs and benefits of defense, independent of the effects of other genes across the genome with the exception of tightly linked loci. Prior to the experiment, 100 seeds (25 of each cyanotype) were germinated, and leaves were phenotyped for the production of hydrogen cyanide, cyanogenic glycosides, and linamarase (i.e., cyanotyped) using Feigl–Anger assays [63]. A total of 49 plants; 11 AcLi, 11 acLi, 16 Acli, and 11 acli were used in the experiment. 

Plants were grown in controlled conditions using a walk-in growth chamber (CMP6050, Conviron, Winnipeg, Canada). Light in the growth chamber was provided by fluorescent and incandescent lamps with a light regime of 15:9 h light:dark under a light intensity of 350 μmol/m^2^/s. The temperature was set to 22 °C during the light period and 17 °C during the dark period. Air humidity was adjusted to 70%. Plants were cultivated in 1 L pots filled with potting soil (PROMIX LP 15, Premium Horticulture, Canada). All plants were fertilized with slow-release fertilizer pellets (Nutricote Total 14-13-13; Plant Products, Leamington, ON, Canada) and randomized across plastic trays. 

### 5.2. Influence of Temperature on the In-Vitro HCN Production

We tested how temperature affected HCN production in-vitro spectrophotometrically. Different volume ratios of linamarin and linamarase were tested. An amount of 10 μL of 10 mM linamarin (Sigma-Aldrich 68264) and 10 μL of 0.2 EU/mL linamarase (LGC Standareds CDX-00012238-100) were added together in Eppendorf tubes and incubated at varying temperatures (5, 10, 15, 20, 25, 30, 35, 40, 45, 50 °C) for two hours in the dark. Thermatron (8200+, Venturedyne, Ltd. Holland, MI, USA) incubators were used to control temperature. The reaction was stopped by putting tubes on ice, and 50 μL H_2_O was added to the reaction. The concentration of HCN liberated was quantified using colorimetric quantification. *Trifolium repens* plants grow along the soil surface and are therefore often exposed to high surface temperatures. The temperature range used sought to mimic and slightly exceed those natural conditions to more fully understand the temperature sensitivity of HCN production.

#### Colorimetric HCN Quantification

HCN was quantified by adding an N-chlorosuccinimide/succinimide oxidizing solution and a chromogenic pyridine/barbituric acid solution to initiate a color-change reaction. After 15 min of incubation at room temperature, the absorbance of each sample was measured using a UV/VIS spectrophotometer (Multiskan GO, Thermo Fisher Scientific, Finland) at 580 nm [64,65].

### 5.3. Effect of Temperature on T. repens Chemical Defense against Herbivores

To examine how temperature influenced the efficacy of chemical defenses in *T. repens* against herbivores, we performed detached leaf herbivory bioassays on all four cyanotypes at different temperatures. We used the grey garden slug (*Deroceras reticulatum*) to quantify herbivore resistance because slugs, including *D. reticulatum*, have been shown to be sensitive to the HCN defense of *T. repens* [32]. Slugs were collected from lawns and gardens in Mississauga, Ontario, Canada and maintained at 23 °C in a terrarium lined with moistened paper towel and provided with cucumber for food before the experiment. The slugs were then starved for 24 h and randomly assigned to treatments. All feeding bioassays were done in Petri dishes, in which one slug was provided with a single detached leaf of each cyanotype. Petri dish bioassays have the benefit of allowing for relatively high replication in highly controlled conditions, but they can be problematic when assaying resistance of defenses that are induced or quickly degraded. However, Petri dish bioassays should be reliable in the case of assessing the resistance of the HCN defense in *T. repens* because the defense is constitutively expressed and the metabolic components are fairly stable, and therefore should be less susceptible to issues of induction or degradation that are inherent to many other types of defenses. Individual slugs were weighed and then placed in separate 9 cm diameter Petri dishes lined with moistened filter paper to keep leaves turgid. All feeding experiments were conducted under two different temperatures (15 °C and 25 °C), with temperature controlled over multiple trials using a Thermotron environmental test chamber. In each Petri dish, leaf material from plants was offered to the slugs for 24 h. Leaves were digitally photographed (Olympus TG-2 iHS Digital Camera, Vietnam) before and after incubation to calculate the leaf area consumed using the Easy Leaf Area software v1.01 [66].

#### 5.3.1. No-Choice Assay

No-choice feeding assays were conducted by placing one leaf of a given cyanotype into a Petri dish and allowing a single slug to feed for 24 h. Two trials at each temperature were conducted for the no-choice assay. In each trial, we used a total of 10 Petri dishes per cyanotype per temperature, each containing one leaf (total N = 80).

#### 5.3.2. Choice Assay

A choice assay was carried out using a cafeteria-style experiment by placing 4 leaves equally spaced in four cardinal directions, one of each cyanotype, into a Petri dish lined with moist filter paper. One slug was placed in the middle of the Petri dish and offered a choice between a leaf from each of the 4 cyanotypes for 24 h. We used a total of 20 Petri dishes, each containing four leaves, 10 dishes for each temperature (total N = 80 leaves).

### 5.4. Freezing Experiment

We examined how freezing affected ATP production and plant physiological responses for each cyanotype. To acclimate the plants prior to freezing, ten plants were selected from each cyanotype, and the temperature was reduced by 2 °C each week for 4 weeks until the temperature reached 14 °C. Plants were then moved to a cold chamber (Biochambers Inc., Winnipeg, Canada) to acclimate at 10/5 °C day/night (15:9 h) for 3 weeks with the lights set to 350 μmol/m^2^/s. 

Plants were subjected to −5 °C for 5 h using the Thermotron. Tissues were collected at five time points: (1) immediately before acclimation (control), (2) immediately before freezing (acclimated), (3) 3 h, (4) 24 h, and (5) 7d after freezing. Plant tissues were subject to multiple analyses to assess freezing damage and autotoxicity. HCN concentration was quantified spectrophotometrically as described above, ATP was quantified using liquid chromatography–tandem mass spectrometry (LCMS/MS), and photosynthetic efficiency was measured using a photosynthesis yield analyzer.

#### 5.4.1. Colorimetric HCN Quantification

To quantify HCN concentrations in all plants at the five time points, HCN was trapped in 1 M NaOH. Two leaflets of uniform size were incubated in 0.1 M citrate buffer and lysed using three freeze/thaw cycles, by freezing the tissues in liquid nitrogen and then thawing the material at room temperature for 15 min. NaOH solutions and the leaf tissues were then incubated for 15 h at 37 °C to initiate a reaction with gaseous HCN, forming NaCN and H_2_O. The concentration of NaCN was then quantified using the colorimetric quantification method described above [64,65].

#### 5.4.2. Measuring the Maximum Photochemical Yield of Photosystem II (Fv/Fm)

To assess the effect of freezing on the photosynthetic activity of white clover, analysis of photosystem II Fv/Fm was measured during the experiment. We measured photosynthetic activity at each of the five time points noted above from a single green leaflet on each whole plant. All plants were dark-acclimated followed by measuring the maximum photochemical yield of photosystem II (Fv/Fm) using a photosynthesis yield analyzer JUNIOR-PAM (Walz Teaching-PAM fluorometer (PAM-200), Heinz Walz GmbH, Effeltrich, Germany) and WinControl-3 software. Three independent measurements were averaged for each plant per time point.

#### 5.4.3. ATP Analysis

To test whether HCN released following freezing leads to autotoxicity by inhibiting ATP synthesis, ATP was quantified for each plant at the five time points described above.

##### Metabolite Extraction

Phosphorylated metabolites were extracted from lyophilized plant tissue for analysis by LCMS/MS using a protocol modified from [67]. All steps were carried out at 4 °C. Briefly, a 10 mg tissue aliquot was extracted by agitation with 250 μL ice-cold 50% (*v*/*v*) acetonitrile containing 10 mM ammonium acetate (pH 9.0). The internal standard 2-deoxy-D-glucose 6-phosphate (DGP; 80 ng) was added to each extract. Following 20 min of agitation, the mixture was centrifuged cold at 16,000× *g* for 10 min. The supernatant was transferred to a fresh tube on ice, and the pellet was extracted a second time and centrifuged as before. After pooling supernatants, the extracts were lyophilized overnight and resuspended in 100 μL of ice-cold 10 mM ammonium acetate (pH 9) and back extracted with 1 volume of chloroform. Following centrifugation, the upper, aqueous phase was filtered through a polytetrafluoroethylene filter (0.2 μm) and diluted with 1 volume of acetonitrile prior to analysis. Calibration curve samples containing 0.1–2 μg of purified ATP and 80 ng internal standard were prepared using the same procedure.

##### Metabolite Quantification

Concentrations of ATP in extracts were measured by LCMS/MS using an Agilent 1290 series II ultra-high pressure liquid chromatograph and a Sciex 4500Qtrap triple quadrupole mass spectrometer. Separation was achieved on a hydrophilic interaction liquid chromatography column (XBridge BEH Amide, 2.5 μm particle size, 2.1 mm × 150 mm; Waters Corporation) fitted with a guard column of the same sorbent. The sample injection volume was 5 μL, and mobile phases consisted of 20 mM ammonium bicarbonate (solvent A) and 80% acetonitrile with 20 mM ammonium bicarbonate (B). A constant flow of 0.5 mL/minute was maintained throughout. The gradient proceeded from 100% B at injection to 84% B by 5 min, followed by a 5 min hold time at 84% B. Solvent B then decreased to 60% over 1 min, and these conditions were maintained for an additional 4 min hold time. At minute 15, a step change to 100% B was applied, and the column was re-equilibrated to initial conditions for 15 min. Mass data were acquired in negative mode with the electrospray ion source configured as follows: ionization −4500 V, source temperature 400 °C, and curtain gas at 20 psi. Multiple reaction monitoring of the target analytes included the following mass transitions: ATP (m/z 506 → 159), declustering potential (DP) −75 V, collision energy (CE) −36 V, cell exit potential (CXP) −1 V; DGP (m/z 243 → 78.9), DP −35 V, CE −62 V, CXP −7 V (collision-induced dissociation, 10 psi nitrogen). Q1 and Q3 operated at unit resolution, and each transition was allotted a dwell time of 50 ms. The instrument was controlled with Analyst software version 1.7.2, and data analysis was performed with Sciex OS version 2.0.0. For quantification, peak areas were normalized to the internal standard, compared to the linear regression obtained from the calibration curve, and converted to pmol∙mg^−1^ dry weight based on sample weight recorded to ±10 μg on an analytical balance.

### 5.5. Statistical Analysis

All statistical analyses were performed in the R computing environment version 4.2.2 [68].

#### 5.5.1. Effect of Temperature on the Production of HCN

To test the effect of temperature on HCN production in vitro, we performed a linear regression, fitting the equation: HCN quantity~Temperature, testing the significance of the effect of temperature using the Anova function in the car package version 3.0-12 [69].

#### 5.5.2. Effect of Temperature on Efficacy of HCN Defense against Herbivores

For all analyses, we first tested whether the residuals from models met assumptions of normality and homogeneity of variance, and we found that a square-root transformation of the leaf area consumed provided the best fit to the data. For the choice assay, we fit the following linear model using the lm function: Leaf Area Consumed~Cyanotype * Temperature. We tested the significance of fixed effects using the Anova function in the car package with type II sums-of-squares (SS). For the no-choice assay, two trials were conducted and analyzed separately using the same model structure as above. We analyzed trials separately due to the larger size and later development stage of the slugs used in trial 2, which resulted in greater feeding and a different response to cyanogenesis. All models originally included slug mass as a covariate, but it was removed in all cases because it was not significant at *p* > 0.1. Finally, in each analysis, we used custom contrasts to compare the mean of cyanogenic (i.e., AcLi) vs acyanogenic (i.e., acli + Acli + acLi) cyanotypes at a given temperature using the emmeans and contrast functions in the emmeans package version 1.8.4-1 [70].

#### 5.5.3. Freezing and Autotoxicity

To examine how freezing affects the physiology of cyanogenic and acyanogenic plants, we first tested the effect of freezing at different time points on HCN production by fitting the linear model: HCN Concentration~ Time Point. We tested the significance of all fixed effects using the Anova function in the car package with type II SS [69]. Mean values at each time point were then compared post hoc using the emmeans function in the emmeans package [70]. Next, we assessed the effect of freezing on photosynthetic activity by comparing Fv/Fm of each cyanotype at multiple time points before and after freezing. We fit the following linear model: Fv/Fm~Cyanotype * Time Point, and tested the significance of fixed effects as described above. Finally, we tested the autotoxicity hypothesis by comparing ATP production of each cyanotype (per mg of plant dry biomass) before and after freezing using the following equation: [ATP]~Cyanotype * Time Point, and tested the significance of fixed effects as described above. We used custom contrasts as described above to compare mean ATP production of cyanogenic and acyanogenic plants within each time point.

## Figures and Tables

**Figure 1 plants-12-01213-f001:**
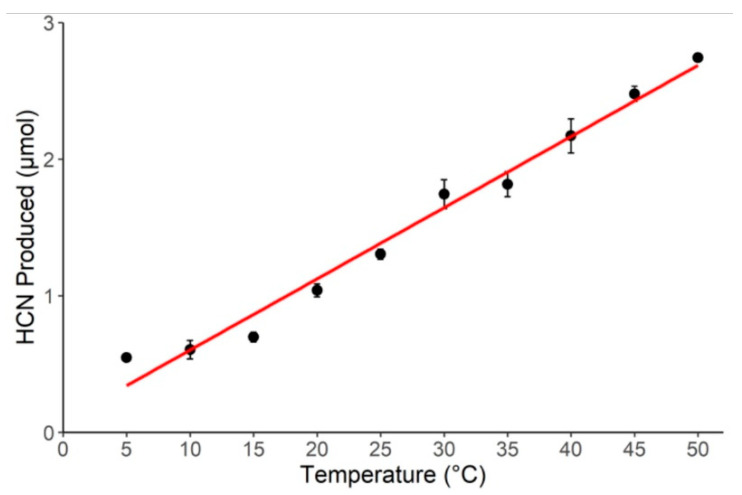
Effect of temperature on the production of HCN. Results are shown as the mean of 3 replicates at each temperature, with error bars representing ± 1 SE (R^2^ = 0.96). Fitting exponential and 2nd-order polynomial curves to the data resulted in lower R^2^ values.

**Figure 2 plants-12-01213-f002:**
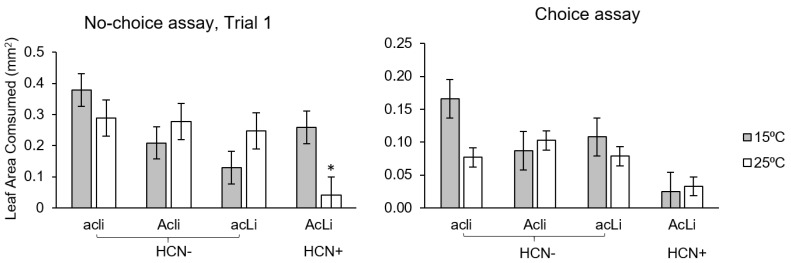
Slug herbivory on *T. repens* cyanotypes at different temperatures (N = 10 plants per cyanotype at each temperature). In the no-choice assay, *D. reticulatum* were offered one leaf of each cyanotype. In the choice assay, leaf material from all four cyanotypes was offered and herbivory was measured as leaf area consumed (mm^2^) over a 24 h period. Bars represent means ± 1 SE. Asterisks (*) represent significant differences (*p* < 0.05) within temperatures between cyanogenic (AcLi) and acyanogenic (average of Acli, acLi, and acli) plants based on custom contrasts (see Methods).

**Figure 3 plants-12-01213-f003:**
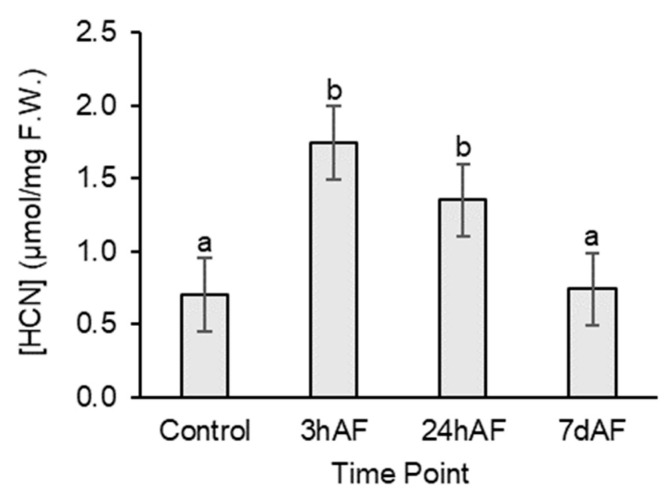
HCN concentration (μmol/mg fresh weight) in leaves of AcLi plants (N = 10) at four time points during the freezing experiment: Control (no freezing), 3 h after freezing (3hAF), 24 h after freezing (24hAF), and 7d after freezing (7dAF). Bars represent means ± 1 SE. Significant differences (*p* < 0.05) among time point means are indicated by lowercase letters after pairwise contrasts.

**Figure 4 plants-12-01213-f004:**
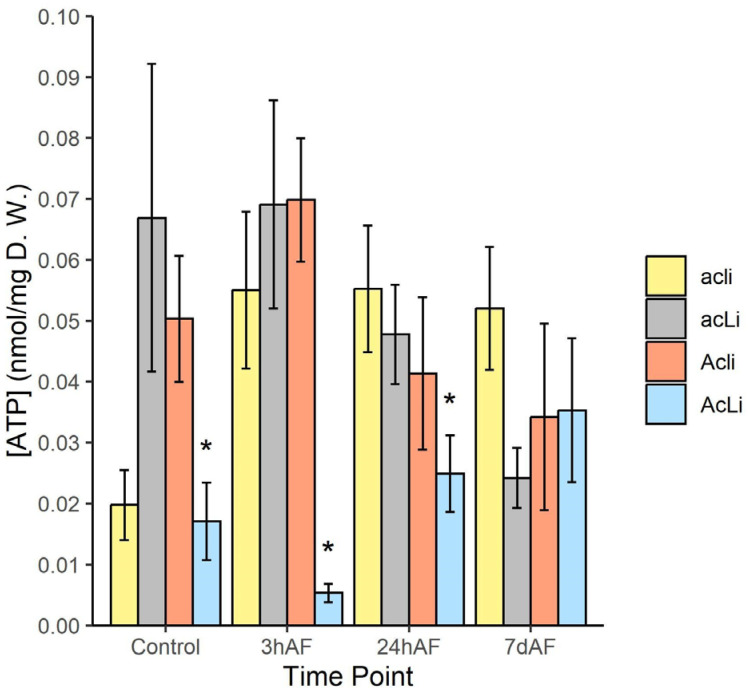
ATP concentrations in nmol/mg dry weight in leaves of four *T. repens* cyanotypes at four time points: Control (no freezing), 3 h after freezing (3hAF), 24 h after freezing (24hAF), and 7d after freezing (7dAF). Asterisks (*) represent significant differences (*p* < 0.05) within time points between cyanogenic (AcLi) and acyanogenic (average of Acli, acLi, and acli) plants based on custom contrasts (see Methods).

**Figure 5 plants-12-01213-f005:**
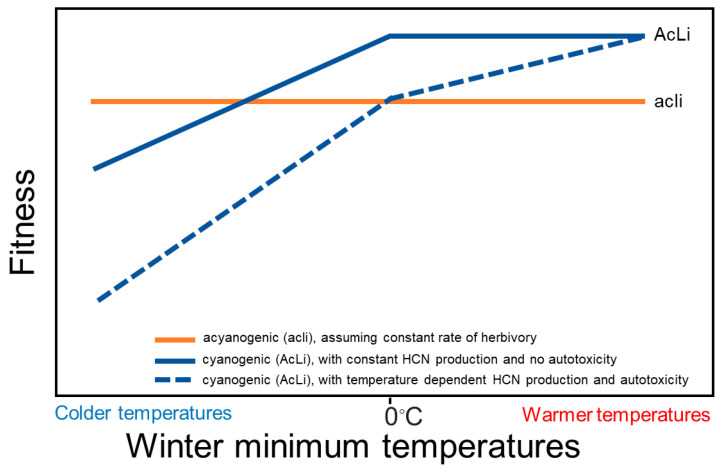
Conceptual model of how the costs and benefits of the hydrogen cyanide (HCN) chemical defense vary with winter minimum temperature in *Trifolium repens*. If there is a constant level of herbivory with variation in temperatures, acyanogenic plants (acli phenotype–orange line) are not expected to show large differences in fitness with variation in temperature. For cyanogenic plants (AcLi phenotype), if HCN expression is not dependent on temperature and if there is no autotoxicity at freezing temperatures (solid blue line), cyanogenic plants are expected to experience higher fitness relative to acyanogenic plants at winter minimum temperatures > 0 °C because of the protection the defense provides against herbivores. AcLi plants are expected to exhibit lower fitness than acyanogenic plants in climates with freezing temperatures because freezing temperatures exacerbate allocation costs of producing cyanogenic glycosides and linamarase [20]. When HCN expression is dependent on temperature and there is freezing-induced autotoxicity (dashed blue line), allocation costs and physiological costs of autotoxicity are compounded at freezing temperatures, and the benefit of the defense increases with temperature > 15 °C. The effects of temperature on the benefits and costs of defense make the evolution of HCN more restrictive. If herbivory also increases with warming temperatures, as predicted by the latitudinal herbivory-defense hypothesis [61], then the conditions for the evolution of HCN are relaxed because the relationship between fitness and temperatures would be negative for acli plants. We display winter minimum temperature on the x-axis because it is freezing temperatures that are hypothesized to induce allocation and autotoxicity costs on plants.

**Table 1 plants-12-01213-t001:** Results from linear models examining how cyanotype, temperature, and their interaction affect herbivory in no-choice and choice assays of *Trifolium repens* leaves.

	**No** **-** **Choice**						**Choice**		
	**Trial 1**			**Trial 2**					
**Effect**	**d.f.**	**F**	** *p* **	**d.f.**	**F**	** *p* **	**d.f.**	**F**	** *p* **
Cyanotype	3	2.386	0.076	3	0.774	0.513	3	1.538	0.212
Temperature	1	0.090	0.765	1	1.578	0.213	1	0.014	0.906
Cyanotype × Temperature	3	2.514	0.065	3	1.416	0.245	3	0.534	0.661

For each response variable, we tested the effects of leaf cyanotype, temperature (15 °C and 25 °C), and the interaction between cyanotype and temperature, for each of two experiments—a no-choice assay, which contained two separate trials, and a choice assay. For each effect, we show the degrees-of-freedom (d.f.) and F- and *p*-values; the degrees-of-freedom for the error term in all assays was 72.

**Table 2 plants-12-01213-t002:** Results from linear models analyzing how ATP concentration in *T. repens* leaves depended on cyanotype, time point, and their interaction.

**Effect**	**d.f.**	**F Value**	** *p* **
Cyanotype	3	2.359	0.075
Time point	3	2.782	**0.044**
Cyanotype × Time point	9	4.171	**<0.001**

We tested the effects of leaf cyanotype, time point of freezing temperature (Control, 3 h, 24 h, and 7 d after freezing), and the interaction between cyanotype and time points. For each effect, we show the degrees-of-freedom (d.f.) and F- and *p*-values. The error degrees of freedom were 123. Significant effects at *p*  <  0.05 are in bold.

## Data Availability

https://doi.org/10.6084/m9.figshare.22242298 (accessed on 8 February 2023).

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
