# Peer review of "Assessing the Benefits and Costs of the Hydrogen Cyanide Antiherbivore Defense in Trifolium repens"

_plants, 2023, doi:10.3390/plants12061213_

Round 1
Reviewer 1 Report
The basic issue in this study was to evaluate the benefits and costs of HCN in defending white clover from an herbivore, the gray slug. The study and analysis was relatively straight forward and examined several variables that have traditionally received relatively little attention, especially considering the amount of attention that has been given the role of HCN in countering herbivory.
My main concerns relate to a general lack of effort to relate the problem to a broader context. All studies were performed in petri dishes and involved work with one host, the white clover and one herbivore, the gray slug. Though Petri dish experiments are commonly used, they are a far cry from issues in the real world, as has been pointed out in the literature.
Clover is attacked by a host of other species, and these others may be as important or more important herbivores than the slug. How useful a model species is the slug for drawing general conclusions about HCN as a protector from herbivores in general? Although considerable attention was paid to varying the temperature regime of the clover, the slugs were kept at a single temperature. What would have been the effect if they had used different temperatures that fell within the range used in the experiment? Would these differ within the range of a season? And would the effects of the adult slugs have shown different effects then? How does the distribution of cyanogenic and acyanogenic plants vary at a local scale?
Do the authors perceive any difficulties in using crosses of Ontario and Louisiana plants in their studies? These plants are widely spaced and presumably adapted to strikingly different environmental conditions, conditions that would seem relevant to the stated environmental concerns of the authors.
In sum, I feel that this was a well-designed study, though I’d have liked to have seen a broader range of experiments, ideally some that followed up insights obtained from the Petri dishes, preferably including some from the field. I feel that the paper would be enhanced by a consideration of such insights.
Author Response
Response to Reviewer 1 report:
1) The basic issue in this study was to evaluate the benefits and costs of HCN in defending white clover from an herbivore, the gray slug. The study and analysis was relatively straight forward and examined several variables that have traditionally received relatively little attention, especially considering the amount of attention that has been given the role of HCN in countering herbivory.
Response: Thank you.
2) My main concerns relate to a general lack of effort to relate the problem to a broader context. All studies were performed in petri dishes and involved work with one host, the white clover and one herbivore, the gray slug. Though Petri dish experiments are commonly used, they are a far cry from issues in the real world, as has been pointed out in the literature.
Response: We appreciate the reviewer’s criticism, and we have tried to put the study in a broader context as described below. But first, some clarification is needed. It is inaccurate to say “all studies were performed in petri dishes, and involved work with one host, the white clover and one herbivore”. Our study included three complementary experiments.
The first experiment was an in vitro experiment to assess the rate of HCN production at different temperatures.
The second experiment was the feeding trial which the author refers to. We generally agree with the limitations of petri dish assays, but they are arguably more reliable in the case of the HCN defense in Trifolium repens because the defense is constitutively expressed, and therefore should be less susceptible to issues of induction or degradation that are inherent to many other types of defenses. We now make this point on Line 417-422 of the Methods, with the addition of the following text:
“Petri dish bioassays have the benefit of allowing for relatively high replication in highly controlled conditions, but they can be problematic when assaying resistance of defenses that are induced or quickly degraded. However, Petri dish bioassays should be reliable in the case of assessing resistance of the HCN defense in Trifolium repens because the defense is constitutively expressed and the metabolic components are fairly stable, and therefore should be less susceptible to issues of induction or degradation that are inherent to many other types of defenses.”
Finally, we performed a third experiment on whole plants in which we exposed them to freezing temperatures and then measured HCN expression, photosynthetic activity and ATP production at multiple time points.
As for a broader context, the introduction already provides a broad introduction to the evolution of plant defense, while we agree the Discussion is fairly narrowly focused on T. repens and the HCN defense. We have therefore added the following text at the end of the Discussion to bring the focus back to larger issues related to the evolution of plant defense:
“More generally, our results contribute to understanding how environmental variation influences the microevolution of plant defenses. Variation in the environment, and particularly variation in resource availability, is well known to influence the macroevolution of defense by modulating the allocation costs of defense [11]. How variation in the environment influences the microevolution of defense is poorly known [13], and the effects of variation in the environment on the benefits of defense have mostly considered variation in the levels and types of herbivory [60]. Our results suggest that variation in edaphic factors not directly related to resource availability or herbivory can modulate both the costs and benefits of defense. We propose that a more complete understanding of how the environment modulates these benefits and costs of defense would improve our ability to predict the evolution of defenses within species.”
3) Clover is attacked by a host of other species, and these others may be as important or more important herbivores than the slug. How useful a model species is the slug for drawing general conclusions about HCN as a protector from herbivores in general?
Response: Slugs have been shown to be sensitive to HCN production in T. repens in previous work, and we now emphasize this point in the text on lines 410-412 of the Methods, with the following addition:
“We used the grey garden slug (Deroceras reticulatum) to quantify herbivore resistance because slugs, including D. reticulatum, have been shown to be sensitive to the HCN defense of T. repens [32].”
4) Although considerable attention was paid to varying the temperature regime of the clover, the slugs were kept at a single temperature. What would have been the effect if they had used different temperatures that fell within the range used in the experiment? Would these differ within the range of a season? And would the effects of the adult slugs have shown different effects then? How does the distribution of cyanogenic and acyanogenic plants vary at a local scale?
Response: We agree it would be interesting to vary temperature and look at the response of herbivores. We chose the temperatures to manipulate expected differences in HCN production based on our in vitro experiment, and to avoid stressing the herbivores or plants. We were limited by both plant tissue and slugs, so extending it past the trials that we did use was infeasible, but could be the focus of future research.
5) Do the authors perceive any difficulties in using crosses of Ontario and Louisiana plants in their studies? These plants are widely spaced and presumably adapted to strikingly different environmental conditions, conditions that would seem relevant to the stated environmental concerns of the authors.
Response: This was not a major concern because the cyanogenesis loci were randomized onto the recombined and segregating Ontario X Louisiana genetic background. So any adaptations associated with the genomes of the Ontario or Louisiana plants aside from cyanogenesis are randomized across Ac/ac and Li/li genotypes. In other words, those effects could not confound the results. We now make this point on lines 372-376:
“This method allowed us to randomize the Ac/ac and Li/li alleles onto a common recombined and segregating genetic background, and therefore to determine the effects of the Ac and Li loci on the costs and benefits of defense, independent of the effects of other genes across the genome with the exception of tightly linked loci.”
6) In sum, I feel that this was a well-designed study, though I’d have liked to have seen a broader range of experiments, ideally some that followed up insights obtained from the Petri dishes, preferably including some from the field. I feel that the paper would be enhanced by a consideration of such insights.
Response: We have added this suggestion to the Discussion on lines 249-251/ 330-332:
Thank you. Please see our response to comment #1. We will also consider the reviewer’s comment in our future work in this system.
Reviewer 2 Report
1. General Comments
The paper presented is very interesting. The work presents a practical problem, characterizes it very well and proposes solutions. The economical approach is well done and gives to the reader a real dimension of the subject.
2. Section by section
2.1. Introduction:
Introduction is very easy to read, very comprehensible and has a lot of references to consolidate the affirmations made.
2.2. Material and Methods:
Material and Methods are, from my point of view, well conducted. The statistical methods are adequate.
2.3. Results:
Results are well presented; graphic component is good, and the presentation is easy to understand.
2.4. Discussion:
Discussion is very well conducted and interesting to read. In my opinion, the discussion concerning “3.2. Freezing and autotoxicity” is a bit deep, maybe it´s hard to read without strong skills in the subject. However, as Plants is a scientific review, I understand the interests of such detail.
3. Suggestions
Legend Fig 5 (Line 313). The mention to “Winter minimum temperatures” must be improved.
4 . Other comments
I like the article and I am very pleased to read it.
Author Response
Thank you, we have revised the legend for Figure 5 accordingly.
Reviewer 3 Report
This manuscript adequately addresses the tradeoffs in plant defense through an elegant model with simple genetics. The manuscript was well-written, the experiment well-designed and the findings clear. I did not find anything to improve upon or change.
Figure 1 covers cyanide production up to 50 degrees Celsius. The biologically relevant temperatures only go up to 30 or 35. I appreciate the overkill, but you can exclude some data points and make the same argument.
Author Response
Response to Reviewer 3:
- This manuscript adequately addresses the tradeoffs in plant defense through an elegant model with simple genetics. The manuscript was well-written, the experiment well-designed and the findings clear. I did not find anything to improve upon or change.
Response: Thank you!
- Figure 1 covers cyanide production up to 50 degrees Celsius. The biologically relevant temperatures only go up to 30 or 35. I appreciate the overkill, but you can exclude some data points and make the same argument.
Response: T. repens grows at the soil surface, so it will be exposed to surface temperatures, which can be quite high during summer. 40°C is quite relevant in this context, and while 50°C may be beyond what is typical, we believe it still provides important context since there is no apparent asymptote in that range. To add clarification, we have added the following text on lines 398-401:
“Trifolium repens plants grow along the soil surface and are therefore often exposed to high surface temperatures. The temperature range used sought to mimic and slightly exceed those natural conditions to more fully understand the temperature sensitivity of HCN production.”